# Energy Consumption Load Forecasting Using a Level-Based Random Forest Classifier

**Yu-Tung Chen [1], Eduardo Piedad Jr. [2] and Cheng-Chien Kuo [1,*]**

[1]  Department of Electrical Engineering, National Taiwan University of Science and Technology, Taipei City 10607, Taiwan

[2]  Department of Electrical Engineering, University of San Jose-Recoletos, Cebu City 6000, Philippines

*  Correspondence: cckuo@mail.ntust.edu.tw; Tel.: +886-02-27333141 (ext. 7710)

**Abstract:** Energy consumers may not know whether their next-hour forecasted load is either high or low based on the actual value predicted from their historical data. A conventional method of level prediction with a pattern recognition approach was performed by first predicting the actual numerical values using typical pattern-based regression models, hen classifying them into pattern levels (e.g., low, average, and high). A proposed prediction with pattern recognition scheme was developed to directly predict the desired levels using simpler classifier models without undergoing regression. The proposed pattern recognition classifier was compared to its regression method using a similar algorithm applied to a real-world energy dataset. A random forest (RF) algorithm which outperformed other widely used machine learning (ML) techniques in previous research was used in both methods. Both schemes used similar parameters for training and testing simulations. After 10-time cross training validation and five averaged repeated runs with random permutation per data splitting, the proposed classifier shows better computation speed and higher classification accuracy than the conventional method. However, when the number of its desired levels increases, its prediction accuracy seems to decrease and approaches the accuracy of the conventional method. The developed energy level prediction, which is computationally inexpensive and has a good classification performance, can serve as an alternative forecasting scheme.

**Keywords:** energy level consumption; pattern recognition; random forest; machine learning; load forecasting; level classification

## 1. Introduction

Energy load forecasting is becoming one of the latest trends due to advancements in energy and power systems and management. As a result, emerging techniques in the field of artificial intelligence (AI) have recently come into play. One particular study reviews various prediction techniques for energy consumption prediction in buildings [1]. Energy regression models are studied in [2]. Machine learning (ML) techniques such as artificial neural networks (ANNs) and support vector machines (SVMs) are employed to predict energy consumption and draw energy-saving mechanisms [3]. Another study reviews the use of a probabilistic approach in load forecasting [4]. Other studies analyze the effectiveness of AI tools applied in smart grid and commercial buildings [5–7]. Most of the studied AI tools focus primarily on actual value forecasting. For example, consumers may not know whether the next-hour forecasted load value based on these models is either high or low. The conventional way is to categorize the forecasted value into reasonable levels, such as low, average, or high, which consumers can understand. This study proposes an alternative method which can be applied when estimated levels instead of actual values are already sufficient for a load forecasting application.

Short-term forecasting of energy consumption load uses the most important historical data ranging from a few hours even up to a number of weeks before the forecasted day. Recently, short-term load forecasting research studies employed advance machine learning such as artificial neural networks [8], fuzzy logic algorithms and wavelet transform techniques integrated in a neural network system [9], and an extreme learning machine [10]. Studies on short-term forecasting also cover various settings such as residential [11], non-residential [12], and micro-grid [13] buildings.

In residential houses, a typical energy consumption forecasting is driven by data generated from humidity and temperature sensors [14]. Occupant behavior assessment can also predict building consumption [15]. A number of research papers study short- and long-term energy consumption both in residential and small commercial establishments. The emergence of algorithms and an increasing computational capability have encouraged the development of different prediction models. Stochastic models can reliably predict the energy consumption of buildings and identify areas of possible energy waste [15–17]. Standard engineering regression and a statistical approach still have good prediction results [1,7,14]. A combination with genetic programming is also effective [18]. Various machine learning tools such as support vector machines and neural network algorithms provide an acceptable energy prediction performance [19,20]. Random forest outperforms other widely used classifiers such as artificial neural networks and support vector machines in energy consumption forecasting [21].

## 2. Machine Learning Methodology

This section introduces the machine learning models and presents their implementation. This covers two parts—the pipeline and implementation of ML models, and the random forest classifier as the ML model used in this study.

### 2.1. Machine Learning Pipeline and Implementation

Figure 1 shows the typical implementation flow of machine learning (ML) algorithms. Two main stages of an ML algorithm are the training and testing phases. First, the training phase creates the ML model using a training dataset based on the chosen ML classifier models. The three most commonly known ML models, namely, artificial neural network (ANN), support vector machine (SVM), and random forest (RF), are employed. The performance validation of the training stage guarantees the general performance of the classifier model and is used to avoid the overfitting issue. Then, verification is performed on the trained model in the testing stage using the testing dataset as input to the trained classifier. This testing dataset is the other partitioned data from the original dataset; therefore, it has identical characteristics to the training dataset. The original dataset is partitioned into 70% and 30% for training and testing, respectively. Performance metrics are used to evaluate both stages. By comparing the performance of both training and testing, any overfitting issue can be determined. It occurs when the training performance is relatively higher than the testing performance.

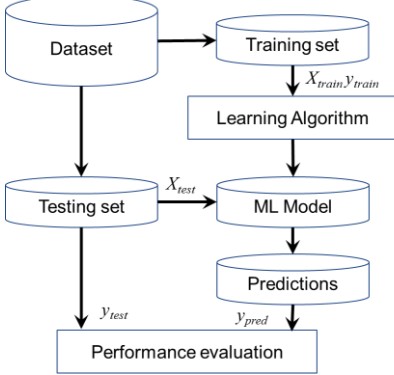

**Figure 1.** Pseudocode of a usual machine learning implementation with training and testing phases, and final evaluation stage.

Algorithm 1 presents the implemented program ML classifier similar to the pseudocode of [21]. A *k* number of times of cross-validation was performed. In this study, 10-time cross-validation was used. This cycle was repeated for another 10 times. The overall average performance and random data shuffling were taken. This verification helped avoid any overfitting issue. This was performed by taking the loss function of random forest present the difference between the training and testing results. For the evaluation of the conventional method, which is a regression-type problem, the root-mean-square error (RMSE) function in Equation (1) is used. F-score, classification accuracy, and confusion matrix are the metrics used for the proposed classifier. F-score accuracy metrics in Equation (2) weigh the significance of both precision and recall performance of the ML model. Precision measures its positive predictive value, whereas recall measures its sensitivity.

---

**Algorithm 1.** Machine Learning Implementation

---

\# Initialization
*In the initialization stage, data pre-processing is performed such as the loading and shuffle-splitting of the dataset into feature X and predictor y, and the importation of the necessary python-based libraries.*
\# Repeat n times the training and testing of the model
for i=1:n
*Shuffle-splitting of dataset into training, validation, and testing datasets*
\# k-time Training Cross-Validation
for j=1:k-time
*Training of the model using an ML algorithm using the training dataset*
*Performance evaluation of the trained model using the validation dataset*
\# Testing the model
*Testing of the trained model using the testing dataset*
*Performance evaluation of the tested model*
\# Display Performance Results
*Compute classification accuracy and F-score*
*Compute classification confusion matrix*

---

Another measure, classification accuracy in (3), was also taken. This metric is the percent of correctly classified levels over the total number of taken levels.

$$\text{RMSE} = \sqrt{\sum_{i=1}^{n} \frac{\left(w^T x(i) - y(i)\right)^2}{n}} \tag{1}$$

$$\text{F score} = \frac{2*\text{Precision} * \text{Recall}}{\text{Precision} * \text{Recall}} \tag{2}$$

$$\text{Accuracy} = \frac{\text{No. of correctly classified energy levels}}{\text{Total number of classified energy levels}} \tag{3}$$

Finally, a confusion matrix normalized between zero and one helps visualize the classification performance of the model. scikit-learn in [22] is an open source platform that provides Python libraries and support. This was used to implement the three ML model classifiers—ANN, SVM, and RF.

*2.2. Random Forest Classifier*

A decision tree was used as the predictive model. The model predicts from the subject observations up to the model decision on which the subject's target value is based. The subject observations are also called branches while subject's target values are also known as leaves. Bagging is a technique of estimated prediction by reducing its variance which is suitable for decision trees [23]. For its regression application, a recursive fit of a similar regression tree was performed to produce bootstrap-sampled versions of training data taking its mean value. For classification, a predicted class was chosen by the majority vote from each committee of trees. Random forest (RF) is a modified bagging that produces a

large collection of independent trees and averages their results [24]. Each of the trees generated from bagging is identically distributed, making it hard to improve other than achieving variance reduction. RF performs the tree-growing process by random input variable selection, thereby improving bagging by the correlation reduction between trees without an excessive variance increase.

## 3. Energy Data Processing

This section presents the implementation of the proposed machine learning classifier using a real energy dataset. The dataset was processed according to the state-of-the-art data class interval method. It was then compared with the conventional forecasting technique.

A 12-month energy dataset of [25] from a large hypermarket was used in this study. An hourly energy consumption collected via a smart metering device and hourly temperature records retrieved from meteorological sensors are shown in Figures 2 and 3, respectively. During the sunny days of the year between June to September, the energy load consumption is relatively high due to the prevalent use of air conditioning units in response to the high temperature.

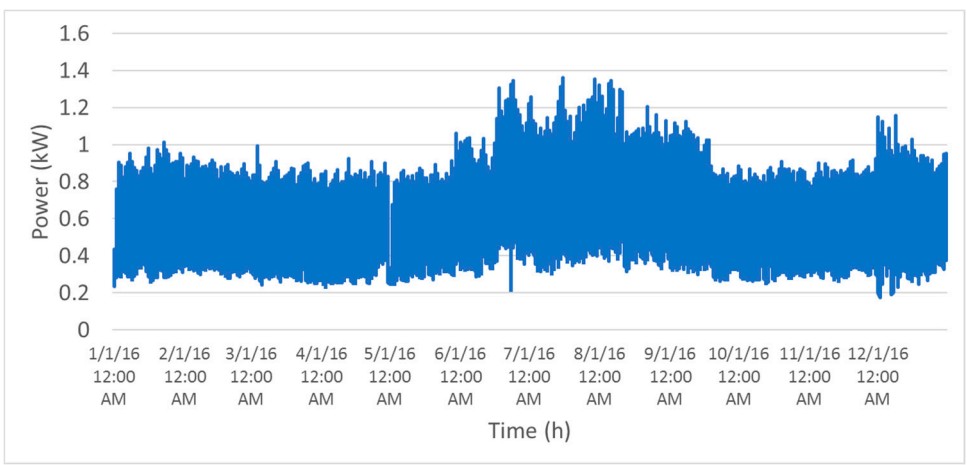

**Figure 2.** Whole-year time series energy consumption data of a commercial entity.

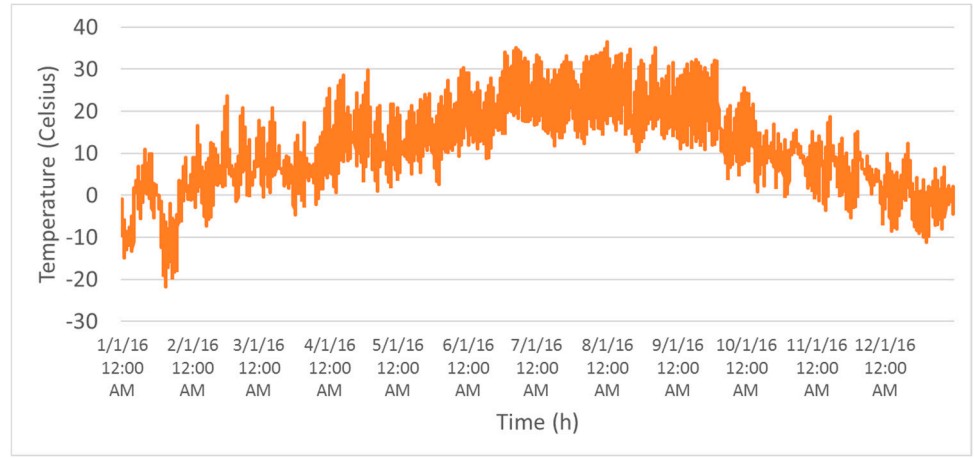

**Figure 3.** Whole-year time series temperature data of a commercial entity.

The conventional method and the new proposed scheme of predicting energy level are shown in Figure 4. The conventional method of energy level prediction is performed by first predicting the actual numerical values using typical regression models and then classifying them into consumer-preferred levels (e.g., low, average, and high). Since the regression model becomes computationally expensive as

its model becomes more complicated, a proposed prediction scheme was developed to directly predict the desired levels using simpler classifier models without undergoing regression.

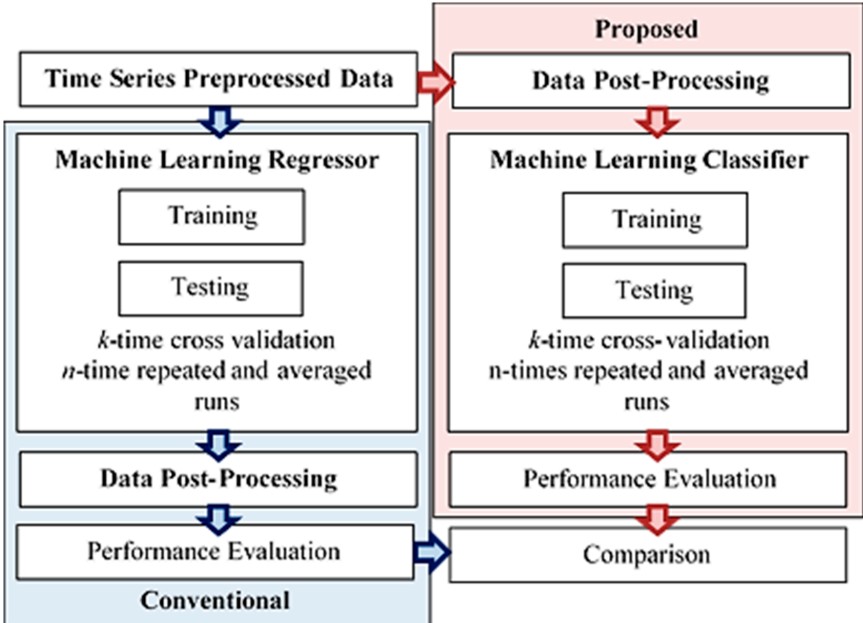

**Figure 4.** Methodology and comparison of the conventional and the proposed time series machine learning classifiers (source: authors' own conception).

In the proposed scheme, the energy consumption values are classified into ordinal bins using a general percentile statistic. Ordinal bin partitioning has an approximately equal number of data points as shown. For example, five bins representing five energy levels (very low, low, mid, high, very high) can be created using five even percentile ranges of the energy consumption data resulting in the [0.174, 0.366), [0.366, 0.634), [0.634, 0.782), [0.782, 0.874), [0.874, 1.36] energy value ranges, respectively. For prediction implementation, energy levels were converted into their respective ordinal values (1, 2, 3, 4, 5) for the machine learning implementation. The dataset contains three energy level cases—three, five and seven classes, as shown in Table 1. The modified dataset can be found in [26]. With these, three prediction cases were conducted using a machine learning random forest classifier.

**Table 1.** Three *n*-level cases for a real dataset.

| n-Level Cases | Interval | Data Points |
|---|---|---|
| 3-level | [0, 0.525) | 2927 |
| | [0.525, 0.807) | 2917 |
| | [0.807, 1.36] | 2940 |
| 5-level | [0, 0.366) | 1755 |
| | [0.366, 0.634) | 1756 |
| | [0.634, 0.7816) | 1759 |
| | [0.7816, 0.874) | 1752 |
| | [0.874, 1.36] | 1762 |
| 7-level | [0, 0.3366) | 1255 |
| | [0.3366, 0.427) | 1249 |
| | [0.427, 0.675) | 1260 |
| | [0.675, 0.771) | 1236 |
| | [0.771, 0.827) | 1271 |
| | [0.827, 0.935) | 1257 |
| | [0.935, 1.36] | 1256 |

## 4. Results and Discussion

This section presents the implementation results of the previous proposed random forest classifier with the previous preprocessed energy data. The results were compared with those of the conventional forecasting classifier.

A brute-force simulation was performed to tune the hyperparameters of both the conventional classifier and the proposed random forest classifier. The training and testing loss function differences of both the conventional and the proposed classifiers are shown in Figure 5. Based on three-level cases, it can be observed that, most of the time, the proposed RF classifier has a lower train–test difference, indicating a better model performance to avoid overfitting compared to the conventional classifier. In addition, the proposed method tends to converge in less than 2% train–test loss function difference as the parameters increase, whereas the conventional method deviates. Furthermore, the average standard deviation on the classification accuracy of the proposed method is lower than the conventional one in all three cases, as shown in Table 2. Accordingly, the lower minimum and maximum standard deviations of the proposed method suggest a more precise prediction than the performance of the conventional method.

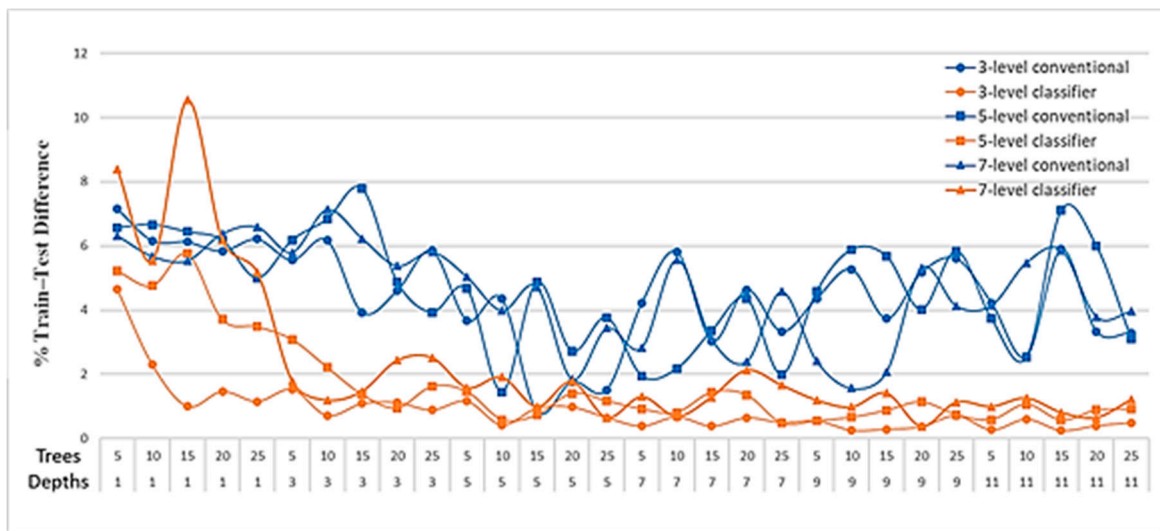

**Figure 5.** Training and testing difference from the loss function graph of the random forest (RF) classifiers.

**Table 2.** Classification performance of the conventional and proposed methods in three cases.

| Classifier Models | | Classification Accuracy | | |
|---|---|---|---|---|
| | | std_min | std_ave | std_max |
| 3-level | conventional | 0.0032 | 0.0106 | 0.9131 |
| | proposed | 0.0012 | 0.0048 | 0.0100 |
| 5-level | conventional | 0.0024 | 0.0093 | 0.0206 |
| | proposed | 0.0023 | 0.0068 | 0.0148 |
| 7-level | conventional | 0.0049 | 0.0100 | 0.0175 |
| | proposed | 0.0033 | 0.0070 | 0.0123 |

The proposed RF classifier tends to perform better with a lower number of energy levels and compared with the conventional method. Based on the F-score in Table 3, the proposed classifier deviates further as the number of levels increases. For example, seven-energy level prediction suggests two times deviation as compared with the three-energy level prediction.

**Table 3.** F-score performance of the proposed method in three cases.

| Proposed | F Score | | | |
|---|---|---|---|---|
| | min | std_ave | max | std |
| 3-level | 0.0054 | 0.6491 | 3.3674 | 0.6388 |
| 5-level | 0.0085 | 1.0993 | 7.1509 | 0.9413 |
| 7-level | 0.0058 | 1.3333 | 5.2442 | 1.1107 |

Parameter simulations of both the conventional and the proposed classifiers in three cases are compared in Figure 6a–c. This was conducted to determine the classification performance and the execution time of both methods as the respective parameters become more complicated. Both classifiers seem to perform better with lower energy levels. Both classifiers converge to a classification accuracy around 90% in three-level prediction in Figure 6a, while reaching around 83% and 75% for five- and seven-level predictions in Figure 6b,c, respectively. However, the proposed classifier is observed to outperform the conventional classifier based on a higher classification accuracy performance and a lower execution time in all three cases. Initially, the execution time of the proposed model takes almost the same time as the conventional one using simpler parameters. With the increasing complexity of the parameters, the former does not change significantly, whereas the latter changes abruptly. It seems that this is due to the fact that the conventional method has a regression model structure which is more complicated than the classification model of the proposed method. The performance of the conventional method approaches that of the proposed method in terms of classification accuracy at the expense of computation time.

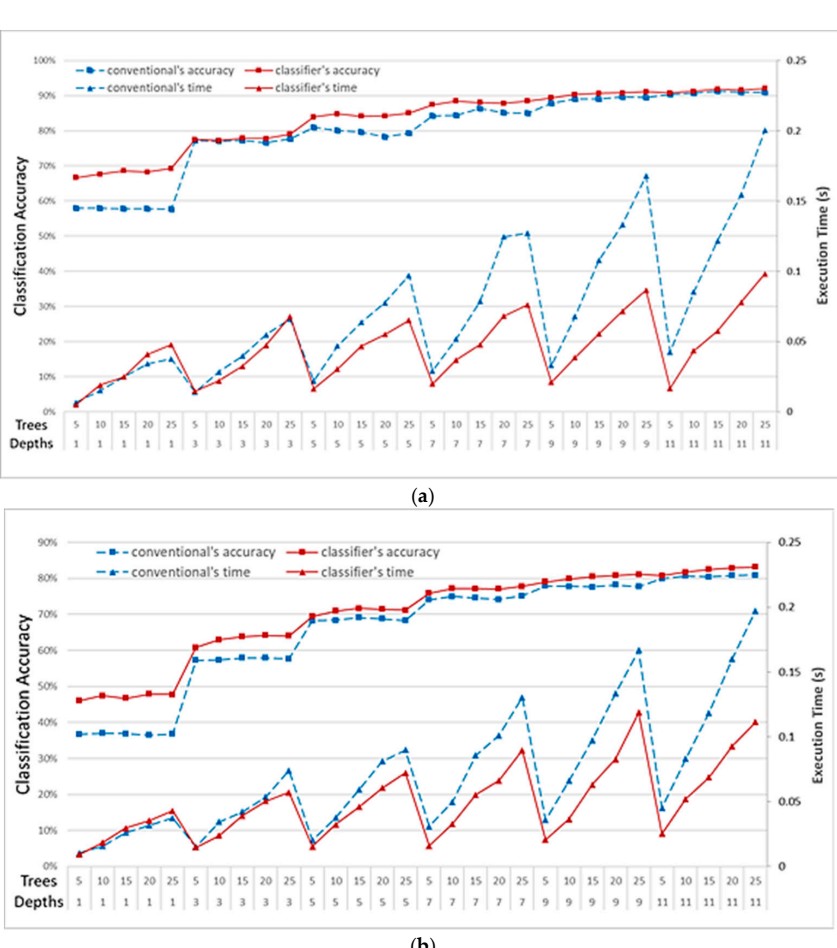

**Figure 6.** *Cont.*

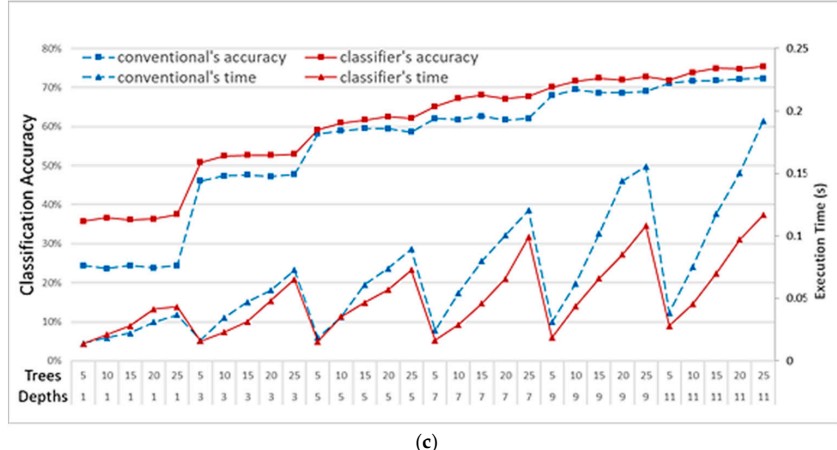

(c)

**Figure 6.** Parameter simulation of three cases, using (**a**) three-energy level, (**b**) five-energy level, and (**c**) seven-energy level prediction of both methods.

## 5. Conclusions

Energy level prediction was performed using a developed random forest classifier. Instead of undergoing regression-based load forecasting from the conventional method, the developed classifier preprocessed the numerical-valued data into levels and then later predicted them using a simpler classification process. Both classifiers perform better with a lower number of energy levels. The developed classifier outperforms the conventional classifier based on its classification accuracy and execution time when simulating 3, 5 and 7 level cases –. However, the performance of the conventional classifier approaches that of the proposed method in terms of classification accuracy but at the expense of computation time. The proposed random forest classifier serves as an alternative to regression-based problems not only for energy consumption forecasting but also for other similar applications. This study was limited to only a single real dataset. Further studies can use other real datasets.

**Author Contributions:** Conceptualization, C.-C.K.; Data curation, Y.-T.C.; Formal analysis, Y.-T.C. and E.P.J.; Investigation, E.P.J.; Methodology, E.P.J. and C.-C.K.; Project administration, C.-C.K.; Software, Y.-T.C.; Supervision, C.-C.K.

**Conflicts of Interest:** The authors declare no conflict of interest.

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
