# Peer review of "Energy Consumption Load Forecasting Using a Level-Based Random Forest Classifier"

_symmetry, doi:10.3390/sym11080956_

Round 1
Reviewer 1 Report
Row 17: Since the regression
R18: becomes
R40: without the first of from the beginning of the row
R59: Smart meter…sentence not finished
R67: of buildings, as well as
R68-69: ...can still predict satisfactorily - phrase consider revising
R75-76: should include introductory text before sub-chapter
R77-78: figure should be announced previously in the text
R131-134: fugures should be announced previously in the text
R141: regressor is underlined with red
R148: becomes
R152: have not has
R153: as shown... where?
R167-170: figure and table not mentioned previously in the text
R185-194: figures and table not mentioned previously in the text
R216: should develop more regarding the Conclusions of the study
R213: consider including more and relevant sources of information (at least 50)
Content must be improved!
Author Response
1. Responses to Comments (Row 17, R18, R40, R59, R67, R68-69, R75-76, R77-78, R131-134, R141, R148, R152, R153, R167-170, R185-194)
- Thanks for the reviewer's kind help. These are checked and applied in the new version of the paper.
2. Responses to Comment (R216)
- Thanks for the reviewer's suggestion. The conclusion of this study adhered to the research outcome. As far as the researchers are concerned, all the findings are remarked in this chapter. However, we will still work on this topic for further informative results.
3. Responses to Comment (R213)
- Thanks for the reviewer's valuable suggestion. The resources related to this new forecasting technique is very limited and not easy to get. However, the main contribution of this study is the novel method which may not always have similar features with other literature. Please see the attached new version of the paper.

Reviewer 2 Report
After I read the manuscript, I have the following remarks (the order is not important):
1. The authors does not follow very clear the general structure for a scientific research paper (article) – introduction, state of the art, methodology, results, conclusions.
2. Chapter 2, 3, 4 can not be started with 2.1. (or worse, with a figure). A short introduction (at least 2-5 sentences are needed).
3. Figure 1, 2, 3 etc. does not mention the source for that data.
4. Figure 4 does not mention the source. If there is the conception of the authors, than this should be stated like this “Source: author’s own conception”.
5. I don’t understand the end of row 59, in the paper – “Smart meter…”
6. Conclusion is very short and at a basic level. Also, they end very steeply (“This random forest classifier can serve as an alternative to regression-based problems not only on energy consumption forecasting but for other similar applications.”). Furthermore, the author does not present the limitations for this research.
Author Response
1. The authors does not follow very clear the general structure for a scientific research paper (article) – introduction, state of the art, methodology, results, conclusions.
- Thanks for the reviewer's comment. The authors carefully followed based on the guidelines of MDPI Journal of Symmetry for a general structure of a scientific research. To make it very clear, we modified the title of Chapters 2 and 3 into Machine Learning Methodology and Energy Data Processing, respectively. Thanks again for your kind help.
2. Chapter 2, 3, 4 can not be started with 2.1. (or worse, with a figure). A short introduction (at least 2-5 sentences are needed).
- Thanks for the reviewer's comment. Short introductions are now added to Chapters 2, 3 and 4.
3. Figure 1, 2, 3 etc. does not mention the source for that data.
- Thanks for the reviewer's comment. Figure 1 is a general machine learning process and we had changed the contents to fit our paper, therefore we did not cite that. The data source for Figs. 2 and 3 are cited in the paragraph. The remaining figures are derived from the data in Figs. 2 and 3.
4. Figure 4 does not mention the source. If there is the conception of the authors, than this should be stated like this “Source: author’s own conception”.
- Thanks for the reviewer's comment. Suggested statement: “(Source: author’s own conception)” is added in the figure caption.
5. I don’t understand the end of row 59, in the paper – “Smart meter…
- Thanks for the reviewer's comment. This is a typo error. It is already removed.
6. Conclusion is very short and at a basic level.
- Thanks for the reviewer's comment. We modified that for more meaningful as the conclusion.
Also, they end very steeply (“This random forest classifier can serve as an alternative to regression-based problems not only on energy consumption forecasting but for other similar applications.”).
- Thanks for the reviewer's comment. The authors modified this sentence into: “The proposed random forest classifier serves as an alternative to regression-based classifier. It can also be used not only for energy consumption forecasting but also for other similar applications.”
Furthermore, the author does not present the limitations for this research.
Thanks for the reviewer's comment. The limitations of this research are implied in chapters 2 and 3. For more completeness, the authors added two sentences in the conclusion stating the limitations of the study. Please see the attached new version of the paper.

Reviewer 3 Report
The text can be better expressed because often leaves some tacit concepts and the English form can also be improved. Figures, Tables and schemes can be resized, generally reduced, because sometimes the caption falls in the following page. The equations seem to be captured by images rather than written within the text. Additionally: half of the page four (4) is a part of a commented programming code that can better be summarized with a scheme and, in any case, matter of a Annex and not of the main text.
The study is correctly designed and technically sounds but the main results of this approach should be tested in other cases of energy consumption load forecasting before generalization of this result. More efforts to describe the methodology and the interpretations of the results should help another researcher to reproduce the results with the same number of pages of the main text by resizing figures and move to an Annex some secondary level details.
Author Response
1. The text can be better expressed because often leaves some tacit concepts and the English form can also be improved. Figures, Tables and schemes can be resized, generally reduced, because sometimes the caption falls in the following page. The equations seem to be captured by images rather than written within the text.
- Thanks for the reviewer's comment. Revised paper already followed the formatting guidelines of MDPI Journal. English form has also improved based on the thorough comments and suggestions from Reviewer 1.
Additionally: half of the page four (4) is a part of a commented programming code that can better be summarized with a scheme and, in any case, matter of a Annex and not of the main text.
- Thanks for the reviewer's comment. Revised based on the guidelines of MDPI, the said programming code is reduced less than half of the page. It cannot be put in Annex because it is only a pseudocode, not the actual programming code, that helps explain the methods of the paper.
2. The study is correctly designed and technically sounds but the main results of this approach should be tested in other cases of energy consumption load forecasting before generalization of this result.
- Thanks for the reviewer's comment. Similar to the response for Reviewer 2, the limitation of this study which is using only a single real dataset is now stated in the conclusion. This makes the generalization of the method limited to said dataset. However, the main contribution of this study is primarily on the developed method. Future research can explore more datasets and more variants of this method.
More efforts to describe the methodology and the interpretations of the results should help another researcher to reproduce the results with the same number of pages of the main text by resizing figures and move to an Annex some secondary level details.
- Thanks for the reviewer's comment. Formatting of figures and tables are now properly addressed based on the guidelines of MDPI journal. Captions of these visual aids are restated to help the readers reproduce the results. English corrections, thoroughly addressed by Reviewer 1, are applied in the new revised paper.
Please see the attached new version of the paper.

Round 2
Reviewer 1 Report
- All the figures and tables must be first announced in the text and only afterwards they should appear;
- Explanations should be included before each table / figure (it is not advisible to include a table right after a figure, with no text and explanations before);
- There must be included an introductory text when chapter 2 begins, before 2.1;
- Figure 5 does not fit in the page correctly;
- Extensive work should be done related to Conclusions and References!
Author Response
1. All the figures and tables must be first announced in the text and only afterwards they should appear;
- Thanks for the reviewer's comment. All of the figures and tables are now announced in the text first before they are shown in the revised manuscript.
2. Explanations should be included before each table / figure (it is not advisible to include a table right after a figure, with no text and explanations before);
- Thanks for the reviewer's comment. These are addressed in the revised manuscript.
3. There must be included an introductory text when chapter 2 begins, before 2.1;
- Thanks for the reviewer's comment. This is now addressed in the revised manuscript.
4. Figure 5 does not fit in the page correctly;
- Thanks for the reviewer's comment. This is also now addressed in the revised manuscript.
5. Extensive work should be done related to Conclusions and References!
- Thanks for the reviewer's comment. Conclusion has to be clear and generalizes the scientific findings in the revised manuscript. The extensive work would be our next research based on this basis. Thanks again for your kind suggestion.

Reviewer 2 Report
I have no further remarks for the revision form of the article.
Author Response
1. I have no further remarks for the revision form of the article.
- Thanks for the reviewer's comment.

Reviewer 3 Report
There are still typing errors like in row 82 where "times" is repeated twice.
Capter 2 starts with paragraph 2.1 and immediately figure 1: it should expectes some text to introduce "Michine Learning" and pseudocode before the scheme representation
Author Response
1. There are still typing errors like in row 82 where "times" is repeated twice.
- Thanks for the reviewer's comment. This is now addressed in the revised manuscript.
2. Capter 2 starts with paragraph 2.1 and immediately figure 1: it should expectes some text to introduce "Michine Learning" and pseudocode before the scheme representation
- Thanks for the reviewer's comment. This is now addressed in the revised manuscript.

Round 3
Reviewer 1 Report
Thank you for the changes made! Good job!